# Age- and Sex-Related Differences in Morbidities of Sexually Transmitted Diseases in Children

**DOI:** 10.3390/children8010040

**Published:** 2021-01-12

**Authors:** Yumika Hino, Nobuoki Eshima, Kira Bacal, Osamu Tokumaru

**Affiliations:** 1Kotake Municipal Hospital, 1191 Katsuno, Kotake-machi, Kurate District, Fukuoka 820-1103, Japan; m08017ye@jichi.ac.jp; 2Center for Educational Outreach and Admissions, Kyoto University, Yoshida-honmachi, Sakyo-ku, Kyoto 606-8501, Japan; eshima.nobuoki.2z@kyoto-u.ac.jp; 3Deputy Head and Phase 2 Director, Medical Programme Directorate, Faculty of Medical and Health Sciences, University of Auckland, 93 Grafton Road, Level 3, Building 532, 1142 Auckland, New Zealand; k.bacal@auckland.ac.nz; 4Department of Physiology, Faculty of Welfare and Health Sciences, Oita University, 700 Dannoharu, Oita 870-1192, Japan

**Keywords:** sexually transmitted diseases, sex-related differences, male-to-female morbidity ratio, *Chlamydia trachomatis*, *Neisseria gonorrhoeae*, condylomata acuminate, genital herpes simplex virus

## Abstract

Sexually transmitted diseases (STDs) are causes of public health burden globally. The purpose of this study is to document age-specific and sex-related changes in the morbidity of four representative STDs in children. Japanese national surveillance data from 1999 to 2017 on morbidities of the following four STDs were analyzed by age and sex: *Neisseria gonorrhoeae* (NG), *Chlamydia trachomatis* (CT), condylomata acuminate (CA), and genital herpes simplex virus (GHSV). The morbidities of males and females in each age group were compared through the male-to-female morbidity (MFM) ratios. The MFM ratios were not different from one in infants, less than one in children, and greater than one after puberty in all four STDs. The reversal of MFM ratio less than 1 to greater than 1 for NG infection was observed between 10–14 and 15–19 year of age, i.e., during the puberty, while that for GHSV infection was observed between 35–39 and 40–44 year of age, i.e., during adulthood. In conclusion, the morbidities of the four STDs were similar between the sexes in infants, and were higher in female children than in male children, while the morbidities in all four diseases were higher in men after puberty.

## 1. Introduction

Sexually transmitted diseases (STDs) are significant causes of public health burden globally. *Chlamydia trachomatis* (CT) is the most commonly reported STD in Japan, followed by *Neisseria gonorrhoeae* (NG) [1]. It has been reported that 48% and 30% of male urethritis is attributable to CT and NG, respectively [2].

NG usually infects the lower genital tract: the cervix in women, and anterior urethra in men. NG can also infect other mucosal surfaces, particularly the pharynx and rectum [3,4,5]. Of note, more than 95% of NG infections in females are asymptomatic and, thus, frequently unrecognized [3,6], which can lead to a significant reservoir of the transmissible bacteria in the population [5,6,7]. NG infection in neonates most often appears in those that are born to mothers with gonococcal genital tract infections, i.e., via vertical transmission, while the transmissions of NG and other STDs in adolescents follow patterns that are similar to those in adults [8].

A better understanding of epidemiological data on STDs, especially age- and sex-related differences in STDs, is vital for the prevention of their transmission. Although the highest incidence of NG is in adolescents and symptomatic patients are reportedly more among males than in females, data on age-specific sex-related differences in infection rates of STDs between both sexes are sparse to the best of our knowledge. Thus, the objective of the present paper is to illustrate age- and sex-related changes in morbidity of NG infection along with CT infection, condylomata acuminata (CA), and genital herpes simplex virus (GHSV) infection. The paper pays particular attention to changes in sex-related differences among infants (younger than one year of age), children (between one and 9–10 years of age) and adolescence (between nine to 10 and 20 years of age) through statistical analyses of the Japanese national surveillance data from 1999 to 2017.

## 2. Materials and Methods

The morbidities of STDs in males and females were compared while using Eshima’s male-to-female morbidity ratios [9,10]

### 2.1. Data

The annual figures of male and female populations of Japan are publicly available online from the Statistics Bureau of Japan, the Ministry of Internal Affairs and Communications [11]. With respect to STDs in Japan, the Ministry of Health, Labour, and Welfare collects reports of patients from approximately one-thousand STD sentinel points on a weekly basis, including NG, CT, CA, and GHSV infections [1]. The criteria for reporting STDs to the sentinel points are as follows; NG: clinically diagnosed as NG infection with the pathogen genetically isolated and identified at laboratory [12], CT: symptomatic with the pathogen genetically isolated and identified at laboratory [13], CA: clinically diagnosed with CA [14], and GHSV: clinically diagnosed with GHSV, excluding recurrence [15]. The data are available through the publicly accessed open database of NIID, which can be used freely without any permission. Data from 1999 to 2017 were included in the following analysis.

### 2.2. Ethics Statement

Ethical approval and signed patient consent forms were not required for our study according to the Guideline for Epidemiological Studies [16], which was established by the Ministry of Health, Labor and Welfare and the Ministry of Education, Culture, Sports, Science and Technology of Japan, in accordance with the World Medical Association’s Declaration of Helsinki and Japan’s Act on the Protection of Personal Information and other related acts. Specifically, (1) all of the data were collected by law and authorized to be utilized for academic purposes [17], and (2) all of the data were deidentified, so individual patients could not be identified.

### 2.3. Statistical Analysis

Age-specific and sex-related differences of reported cases of NG, CT, CA, and GHSV were statistically analyzed for their relative risks. This is a cross-sectional study, thus ages denote the ages in each year. Morbidities of males and females in each age group were compared through the male-to-female morbidity (MFM) ratios [9,10], with statistics that are similar to ones used by Green [18] and Reller et al. [19]. Because the present sampling is based on the data reported from the sentinel points, the sampling is viewed as a Poisson sampling. The morbidities (symptomatic incidence) of males and females at a specific time, $p_{M}$ and $p_{F}$, cannot be estimated from the observational patient data.

Let $\pi_{M}$ and $\pi_{F}$ be the probabilities that male and female patients in the age group visit the sentinel points, respectively. From the present sampling from the sentinel points, the ratio $\gamma =\frac{\pi_{M}p_{M}}{\pi_{F}p_{F}}$ can be estimated by maximum likelihood estimator γ^= nMNMnFNF=nMNFnFNM, where $N_{M}$ and $N_{F}$ are the subpopulations of males and females in an age group in Japanese population; i.e., fixed values, and $n_{M}$, and $n_{F}$ are the random variables that describe the accumulated numbers of male and female patients through the observation period. Let $\delta =\frac{N_{M}}{N_{F}}$ be the ratio of the male population to the female population. Despite that annual variation in $N_{M}$ and $N_{F}$, δ is almost constant through the years for every age group in Japan, e.g., for under 20 year-old δ≈1.05. We used the averages of the ratios δ through 1999 to 2017 (Table 1). Subsequently, estimate $\hat{\gamma }$ is expressed by $\hat{\gamma }=\frac{n_{M}}{n_{F}}\left(\frac{1}{\delta }\right)$. The ratio is referred to as the apparent MFM ratio. If $\frac{\pi_{M}}{\pi_{F}}=1$, then, $\gamma =\frac{p_{M}}{p_{F}}$ is the true MFM ratio. For large $n_{M}$ and $n_{F}$, $\log\hat{\gamma }$ is asymptotically normally distributed with mean logγ and variance $\frac{1}{n_{M}}+\frac{1}{n_{F}}$.^14^ The 95% confidence intervals (95%CI) of MFM ratios were also given, as follows: $\exp\left(\log\hat{\gamma }-1.96\sqrt{\frac{1}{n_{M}}+\frac{1}{n_{F}}}\right)<\gamma <exp\left(\log\hat{\gamma }+1.96\sqrt{\frac{1}{n_{M}}+\frac{1}{n_{F}}}\right)$ [20]. If $n_{M}$ or $n_{F}$ is small, i.e., less than 5, 95%CI is made, as follows. Given $n_{M}+n_{F}$, the distribution of $n_{M}$ is a binomial distribution with success probability $q=\frac{p_{M}}{p_{M}+p_{F}}$ and the number of trials $n_{M}+n_{F}$. Let $F\left(x|n_{M}+n_{F},q\right)$ be the distribution function. Afterwards, the lower and the upper bounds of 95%CI of q, $q_{lower}$ and $q_{upper}$ are calculated by solving $F\left(n_{M}|n_{M}+n_{F},q\right)=0.975$ and $F\left(n_{M}|n_{M}+n_{F},q\right)=0.025$ with respect to q, respectively. From this, 95%CI of γ is given by $\frac{q_{lower}}{1-q_{lower}}\left(\frac{1}{\delta }\right)<\gamma <\frac{q_{upper}}{1-q_{upper}}\left(\frac{1}{\delta }\right)$. The significance level for testing the null hypothesis $H_{0}:\gamma =1$ vs. the alternative hypothesis $H_{1}:\gamma \neq 1$ was set at 0.05.

The MFM ratios were calculated by a spreadsheet software Excel Ver.16.16.26 (Microsoft, Redmond, WA, USA), and statistical analyses were conducted with a statistical software R ver. 3.6.3 (The R Foundation for Statistical Computing, Vienna, Austria).

## 3. Results

Table 1 shows the four data sets for the total number of reported cases from 1999 to 2017, and the estimated MFM ratios of the STDs that are classified by age and sex. Figure 1 also illustrates the MFM ratios and 95%CI.

### 3.1. Neisseria Gonorrhoeae

The MFM ratios for NG infection were less than one under 15 years old, except infants, i.e., 0 year-old children (*p* < 0.001), and greater than one for those aged 15 years or older (*p* < 0.001; Table 1, Figure 1A). In other words, female children and adolescents that were younger than 15 years were reported as infected significantly more often than male children, while thereafter NG infections in males were more frequently reported.

### 3.2. Chlamydia Trachomatis

The MFM ratios for CT infections were not statistically significant in age groups that were younger than 10 years old (Table 1, Figure 1B). In 10–29 years old, MFM ratios were less than one (*p* < 0.001); i.e., in adolescents and adults younger than 30 years of age, females were reported as being infected significantly more frequently than males. By contrast, in those over 30 years of age, the MFM ratios were greater than one, i.e., males were more frequently reported to be infected with CT than females.

### 3.3. Condylomata Acuminata

There was no significant sex-related difference in CA in those 0–4 years old; however, 5–9 year old boys were more frequently reported to suffer from CA than girls, an MFM ratio of 3.33 (95% CI, 1.52–7.31). The MFM ratios for CA were less than one in 10–24 years old, then greater than one over 25 years old; i.e., female patients were more frequently reported with CA in adolescents and adults younger than 25 years old, while the relationship was reversed thereafter (Table 1, Figure 1C).

### 3.4. Genital Herpes Simplex Virus

There was no significant sex-related difference in the GHSV infection patterns in 0 year-old infants. MFM ratios for GHSV infection were statistically less than one in 1–39 years old and greater than 1 over 40 years old, i.e., female children, adolescents, and adults younger than 40 years old were reported as infected with GHSV significantly more often than males (Table 1, Figure 1D), but GHSV was more often reported in men thereafter.

## 4. Discussion

The present paper focused on the age-specific changes in male-to-female relative risks (morbidities) of the following four STDs—NG, CT, CA, and GHSV—especially in children. According to the statistical analysis that is shown in Table 1, the MFM ratios in infants were not statistically significant for all four STDs, indicating that the risks of infection with those STDs are equal for both male and female infants. These findings suggest that most of the infections in this age group would be vertically transmitted, with male and female infants being equally likely to be infected from their mothers [8].

In NG infection, the MFM ratios were less than one in those between 1–14 years of age, i.e., the relative risks of infection were higher for prepubertal females over males, as shown in Table 1A and Figure 1A. The same phenomenon was also observed with GHSV infection in this age group, although GHSV rates remained less than one until patients reached the age of 39 years. In CT infection and CA, the MFM ratios were less than one in those aged 10–29 and 10–24 years, respectively.

To the best of our knowledge, the present paper is the first to report the findings that female-dominance in reported morbidities occurs by 10 years of age in all four STDs examined, with specific patterns of timing varying among the specific pathogens. At some point thereafter, the MFM ratios for all four STDs switched to greater than one, so that, by some point in adulthood, the male patients were more frequently reported to be infected than females.

The observed age-specific and sex-related changes in morbidity might be due to the biology of the pathogens, changes in physiological barriers to pathogens during adolescence, and behavior patterns during and after adolescence of both sexes. For example, during puberty in girls, an increasing level of estrogen thickens vaginal epithelium and causes vaginal pH to acidify, which increases the resistance to microorganisms, including NG [21]. Given that such increases in resistance occur more in females than males, it is reasonable to expect the MFM ratio of NG to increase during and after puberty.

The observed reversal of MFM ratios (from less than one to greater than one) might also reflect the differences in sexual behavior between the sexes. A meta-analysis indicated that risky sexual behaviors were more common in adolescent girls than boys, which could be associated with a decreased MFM ratio during puberty [22].

The MFM ratio for CA was greater than one in the 5–9-year age group. This is in keeping with previous reports [9,10], where the MFM ratios in common pediatric infectious diseases among Japanese children were generally greater than 1. Interestingly, in the present study, MFM ratios that were greater than 1 in childhood were only observed for CA. All of the STDs studied had periods before or during adolescence when their MFM ratios were less than 1. This has not previously been reported and it suggests differences in the mode of transmission for STD infections vs. common pediatric infections.

With respect to NG infection, one possible explanation would be the fact that NG survives in intracorporeal environments and infects the columnar epithelium. Prepubertal female children could be more susceptible to gonococcal vulvovaginitis for several reasons, including immature vaginal barrier functions, due to the lack of estrogen [21,23,24]. NG is an exclusive human pathogen [5], and humans serve as the sole natural host [7]. Thus, after adolescence, the usual mode of transmission of NG is sexual contact [8]. However, transmission via communal baths, towels or fabric and caregivers’ hands has previously been reported as a means in prepubertal children [25]. Other studies suggest that MFM ratio less than one in prepubertal age are indicative of sexual abuse of female children by male adults [26,27,28]. Because the present study found no difference in the morbidities of CT infection and CA in 1–9 year of age between boys and girls, female dominance in NG infection in the present study could not be necessarily indicative of sexual abuse in this age group.

The MFM ratios reversed from less than one to greater than one after adolescence for all four STDs. Because the four infections are all sexually transmitted, it might be expected that they would show similar patterns of infection and transmission. However, the present study showed that the reversals from female predominance to male occurred at quite different ages: at 15–19 year old for NG, 25–29 years old for CA, 30–34 years old for CT, and 40–44 years old for GHSV. The reversal of MFM ratio occurred earliest in NG infections. This may be due to the asymptomatic nature of NG infections in ≥95% of women, while ~90% of men with NG infections are symptomatic [3]. This large difference in symptomatology could account for the apparent MFM ratio of 1.4 in 15–19 years, as it might help to mask the true numbers of infected females; i.e., 90%/5% = 18; the true MFM ratio could be less than one.

In the case of sexually transmitted diseases, medical care-seeking behavior could differ between sexes and among age groups [29], which might be related to stigma regarding STDs among female adolescents [30]. Another survey reported that mean days delay in seeking treatment was 11 days among adolescent youths, reason for which included lack of knowledge about symptoms and inconvenience due to clinic hours [31]. However, there are no data available on the probabilities of care-seeking behavior to the best of our knowledge. Therefore, the authors have tried another approach while using male-to-female immunization ratios of measles-rubella combination vaccine as a proxy. In 2012, the male-to-female immunization ratios of measles-rubella combination vaccine were 0.991 (95% CI 0.987–995) for one year old, 1.00 (0.999–1.01) for 5–6 years, 0.989 (0.985–0.992) for 13 years, and 0.976 (0.972–0.980) for 18 years of age, indicating virtual equality in vaccination rates [32]. Thus, the authors speculate that a sex- and age-based bias in the probability of seeking medical care is unlikely at least under 18-year-old, i.e., from infants to adolescents.

The description that most of the NG infections in females are asymptomatic and, thus, unrecognized [3,6] could lead to an impression that females are not more frequently infected than males in any age groups. The findings of this paper, on the contrary, would indicate that female children and adolescents are more frequently infected than males. It might imply a presence of unknown mechanism of sex-related difference in NG infection, by which a significant reservoir of the transmissible bacteria in the population [5,6,7] could be explained. It might also be indicative of an unrecognized difference in behaviors between boys and girls. The periods with female-dominance depended on pathogens, while all off the pathogens studied showed periods with MFM ratios less than one. The former would reflect pathogen-specific aspect of sex- and age-related differences in morbidity, while the latter would reflect the behavioral pattern of children and adolescents that are specific to sexes, both of which are of clinical importance in pediatrics.

Some limitations of the present study should be noted. First, the present study was based on reported data on STDs to the national surveillance system. Not all of the infections are reported, and patterns of non-reporting may vary between the sexes. This could lead to bias on sex-related differences of morbidities. It is also possible that reporting rates are influenced by age as well as sex, although no quantitative evidence of such influences in Japan was not found, as described above. Second, the present study only analyzed data of symptomatic cases. The omission of asymptomatic cases might lead to biased results between the sexes. That said, we would expect that the high proportion of asymptomatic NG cases in females would lead to under-reporting of female cases. This would then strengthen the current observation of MFM ratio less than one with NG infection [3] and suggest the current finding is more likely to be accurate. Thirdly, it should be noted that the present paper is not a report on the burden of the diseases. It focuses on a comparison of morbidities between both of the sexes with reference to age. Fourth, the present estimation was carried out under an assumption that there was no year-effect during the observational period (1999–2017). It is true that the numbers of cases varied from year-to-year. However, the authors confirmed that the MFM ratios rarely varied. Because the numbers of cases under 14-year-old were too small to follow on a year-to-year basis, the data had to be accumulated to reveal possible sex-related differences in those age groups. Fifth, there could be cases of recurrence or co-infection. It is true that co-infection rate of CT was approximately 20–30% in patients with NG [2,33]. Because the database of NIID did not provide such information, these factors were not included in the present study.

In conclusion, this study illustrated age-specific and sex-related changes in the morbidity of NG, CT, CA, and GHSV infections through statistical analyses of the national surveillance data in Japan. The authors illustrated no sex-related difference in infants, female-dominance in morbidity by puberty, followed by male-dominance thereafter in all of the STDs under review.

## Figures and Tables

**Figure 1 children-08-00040-f001:**
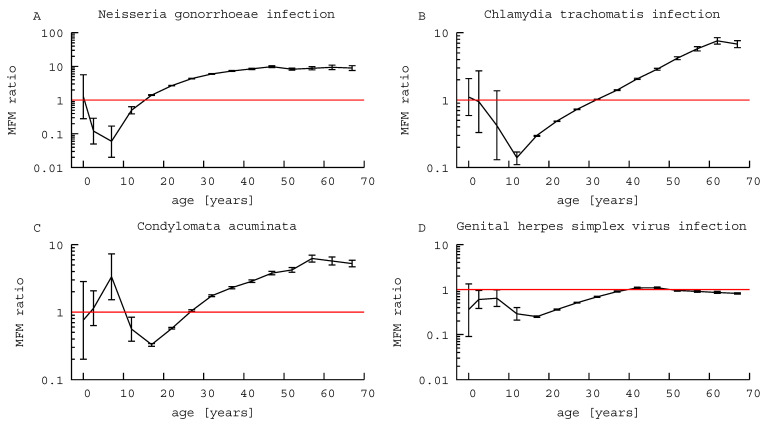
Male-to-female morbidity ratios of sexually transmitted diseases. (**A**) *Neisseria gonorrhoeae* infection. (**B**) *Chlamydia trachomatis* infection. (**C**) Condylomata acuminata. (**D**) Genital herpes simplex virus infection. MFM ratio: male-to-female morbidity ratio. Error bars: 95% confidence intervals.

**Table 1 children-08-00040-t001:** Data of reported cases infected with sexually transmitted diseases, and male-to-female morbidity ratios in age groups.

Age Group	[year]	0	1–4	5–9	10–14	15–19	20–24	25–29	30–34	35–39	40–44	45–49	50–54	55–59	60–64	65–
Male-to-female population ratio δ		1.053	1.050	1.050	1.051	1.051	1.048	1.031	1.025	1.022	1.016	1.006	0.993	0.972	0.943	0.697
(A) *Neisseria gonorrhoeae* (NG)	male	4	5	4	96	11980	39044	40405	32655	24074	15953	9776	6004	3243	1612	1036
	female	3	41	60	182	8079	13844	8989	5345	3234	1869	995	729	379	182	158
	MFM ratio	1.27	**0.12**	**0.06**	**0.50**	1.41	2.68	4.36	5.96	7.28	8.42	9.77	8.29	8.81	9.39	9.03
	(95%CI)	(0.43–9.56)	(0.05–0.31)	(0.03–0.18)	(0.39–0.64)	(1.37–1.45)	(2.63–2.74)	(4.26–4.46)	(5.79–6.14)	(7.02–7.55)	(8.01–8.82)	(9.15–10.43)	(7.68–8.95)	(7.92–9.79)	(8.06–10.95)	(7.84–10.39)
	*p*	0.539	0.000	0.000	0.000	0.000	0.000	0.000	0.000	0.000	0.000	0.000	0.000	0.000	0.000	0.000
(B) *Chlamydia Trachomatis* (CT)	male	21	7	4	115	16505	52428	52698	42404	31275	21374	13539	8562	4743	2575	1601
	female	18	7	9	792	52499	102119	69791	40670	21798	10211	4697	2059	843	360	322
	MFM ratio	1.11	0.95	0.42	**0.14**	**0.30**	**0.49**	**0.73**	1.02	1.40	2.06	2.87	4.19	5.79	7.58	7.13
	(95%CI)	(0.59–2.08)	(0.33–2.72)	(0.17–1.73)	(0.11–0.17)	(0.29–0.30)	(0.48–0.49)	(0.72–0.74)	(1.00–1.03)	(1.38–1.43)	(2.01–2.11)	(2.77–2.96)	(3.99–4.39)	(5.38–6.23)	(6.79–8.47)	(6.33–8.04)
	*p*	0.750	0.926	0.133	0.000	0.000	0.000	0.000	0.012	0.000	0.000	0.000	0.000	0.000	0.000	0.000
(C) Condylomata Acuminata (CA)	male	4	24	28	35	1972	8422	10,624	10,441	8463	6533	4641	3001	1944	1310	1465
	female	5	20	8	60	5766	14,044	9813	5847	3609	2244	1219	714	321	242	380
	MFM ratio	0.76	1.14	3.33	**0.56**	**0.33**	**0.57**	1.05	1.74	2.29	2.87	3.79	4.23	6.23	5.74	5.53
	(95%CI)	(0.28–4.57)	(0.63–2.07)	(1.52–7.31)	(0.37–0.84)	(0.31–0.34)	(0.56–0.59)	(1.02–1.08)	(1.69–1.80)	(2.21–2.38)	(2.73–3.01)	(3.55–4.03)	(3.90–4.59)	(5.54–7.01)	(5.00–6.58)	(4.94–6.19)
	*p*	0.611	0.658	0.003	0.006	0.000	0.000	0.001	0.000	0.000	0.000	0.000	0.000	0.000	0.000	0.000
(D) Genital Herpes Simplex Virus (GHSC)	male	3	29	35	52	1467	6553	9333	9894	9165	7920	5980	4925	3599	2697	4496
	female	8	46	52	170	5581	17569	17625	13952	9979	7157	5381	5202	4077	3308	7445
	MFM ratio	0.36	**0.60**	**0.64**	**0.29**	**0.25**	**0.36**	**0.51**	**0.69**	**0.90**	1.09	1.10	0.95	0.91	0.86	0.87
	(95%CI)	(0.13–1.69)	(0.38–0.96)	(0.42–0.98)	(0.21–0.40)	(0.24–0.26)	(0.35–0.37)	(0.50–0.53)	(0.67–0.71)	(0.87–0.92)	(1.06–1.12)	(1.06–1.15)	(0.92–0.99)	(0.87–0.95)	(0.82–0.91)	(0.83–0.90)
	*p*	0.611	0.032	0.042	0.000	0.000	0.000	0.000	0.000	0.000	0.000	0.000	0.013	0.000	0.000	0.000

MFM ratio: male-to-female morbidity ratio (boldface indicates ratios significantly less than 1); 95%CI: 95% confidence interval.

## Data Availability

Publicly available datasets were analyzed in this study. This data can be found here: https://www.mhlw.go.jp/topics/2005/04/tp0411-1.html.

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
