# Peer review of "Age- and Sex-Related Differences in Morbidities of Sexually Transmitted Diseases in Children"

_children, 2021, doi:10.3390/children8010040_

Round 1

Reviewer 1 Report

This is a difficult work with some biases. However the Authors listed all of them. The work can be considered the first level for successive researches. The topic is interesting and quite original.

Author Response

The authors sincerely appreciate the thorough review and encouraging comment by the Reviewer 1.

Reviewer 2 Report

The authors present an article titled “Age- and sex-related differences in morbidities of 3 sexually transmitted diseases in children”. This study present age-specific and sex-related changes in mobility of four representative STDs in children.  This paper is interesting, the results are clear and the discussion is adequate.

Author Response

The authors sincerely appreciate the thorough review and encouraging comment by the Reviewer.

Reviewer 3 Report

The topic involves innovative perspectives and it has value for prevention of STDs. However, I think that authors could strengthen the introduction by describing in more detail why this scientific knowledge was needed. Now this has been mentioned in rows 44-45 but quite superficially. In addition, authors could describe in more detail what is the clinical relevance of this topic. The literature available about the topic has been explored well and references are relevant. 

Authors have defined key concepts in rows 48-49. However, this definition (infant = younger than 1 year, adolescence = 9-20 years) is lacking years between 1 and 9. In addition, this definition does not include concept "child" which is used when reporting the results. Hence, I suggest that authors would check that definition regarding key concepts (age groups) and use it use them systematically throughout the text.

The objective and scope have been defined well and the topic is strongly related to understanding epidemiological data on STDs, especially age- and sex-related differences regarding infections. Since this publication (Children) focuses on sharing science relevant to children´s health, it would be a great value for this manuscript if authors could link the objective of this manuscript more strongly to infants, children, and adolescents. This comment applies to the presentation of results as well.

The study has been planned and implemented applying research methods and selection of those methods is justified. However, it would be good if the authors could name research method used at the beginning of chapter 2.

The conclusions are justified based on results. However, it would be a great value for this manuscript if the authors could describe what are clinical implications of this study. Furthermore, what are recommendations for clinical practice and policy makers based on this study.

To sum up, the structure of the article is consistent. Clarity and readability are good.

Author Response

The topic involves innovative perspectives and it has value for prevention of STDs. However, I think that authors could strengthen the introduction by describing in more detail why this scientific knowledge was needed. Now this has been mentioned in rows 44-45 but quite superficially. In addition, authors could describe in more detail what is the clinical relevance of this topic. The literature available about the topic has been explored well and references are relevant. 
=> The reason why the topic of the present study is necessary was the fact that data on age-and sex-related differences in infection of STDs were sparse as far as the authors surveyed. Description has been added to the introduction section (lines 39–41). A paragraph was also added to the discussion section regarding the clinical relevance of the study (lines 198–206).

Authors have defined key concepts in rows 48-49. However, this definition (infant = younger than 1 year, adolescence = 9-20 years) is lacking years between 1 and 9. In addition, this definition does not include concept "child" which is used when reporting the results. Hence, I suggest that authors would check that definition regarding key concepts (age groups) and use it use them systematically throughout the text.
=> The authors have defined “children” as aged 1 to 9 years. The words “infants”, ”children” and ”adolescents” are used to indicate specific age groups throughout the manuscripts as suggested.

The objective and scope have been defined well and the topic is strongly related to understanding epidemiological data on STDs, especially age- and sex-related differences regarding infections. Since this publication (Children) focuses on sharing science relevant to children´s health, it would be a great value for this manuscript if authors could link the objective of this manuscript more strongly to infants, children, and adolescents. This comment applies to the presentation of results as well.
=> The description in the results section was modified using the above definition of age groups. Difference in morbidity between both sexes in pediatric patients are explained focusing on pediatric age groups.

The study has been planned and implemented applying research methods and selection of those methods is justified. However, it would be good if the authors could name research method used at the beginning of chapter 2.
=> The research method was introduced at the beginning of the methods section as indicated. The method was named after the statistician who came up with the use of MFM ratio.

The conclusions are justified based on results. However, it would be a great value for this manuscript if the authors could describe what are clinical implications of this study. Furthermore, what are recommendations for clinical practice and policy makers based on this study.
=> Clinical implications of the present study was added to the discussion section as suggested (lines 198–206).

To sum up, the structure of the article is consistent. Clarity and readability are good.
=> The authors sincerely appreciate the thorough review and constructive comments by the Reviewer 3.